# Hydrochlorothiazide/Losartan Potassium Tablet Prepared by Direct Compression

**DOI:** 10.3390/pharmaceutics14081741

**Published:** 2022-08-21

**Authors:** Qiuhua Luo, Qianying Zhang, Puxiu Wang

**Affiliations:** 1Department of Pharmacy, The First Affiliated Hospital of China Medical University, Shenyang 110001, China; 2School of Pharmacy, China Medical University, Shenyang 110122, China; 3Department of Pharmaceutics, College of Pharmacy, Shenyang Pharmaceutical University, Shenyang 110016, China

**Keywords:** direct compression, compression and mechanical properties, D-optimal mixture experimental design, dissolution rate, bioavailibility

## Abstract

Hydrochlorothiazide (HCTZ)/losartan potassium (LOS-K) was used as a model drug to prepare compound tablets through the investigation of the compression and mechanical properties of mixed powders to determine the formulation and preparation factors, followed by D-optimal mixture experimental design to optimize the final parameters. The type and amount of lactose monohydrate (SuperTab^®^14SD, 19.53–26.91%), microcrystalline cellulose (MCC PH102, 32.86–43.31%), pre-gelatinized starch (Starch-1500, 10.96–15.91%), and magnesium stearate (0.7%) were determined according to the compressive work, stress relaxation curves, and Py value. Then, the compression mechanism of the mixed powder was investigated by the Kawakita equation, Shapiro equation, and Heckel analysis, and the mixed powder was classified as a Class-II powder. The compaction pressure (150–300 MPa) and tableting speed (1200–2400 Tab/h) were recommended. A D-optimal mixture experimental design was utilized to select the optimal formulation (No 1, 26.027% lactose monohydrate, 32.811% MCC PH102, and 15.462% pregelatinized starch) according to the drug dissolution rate, using Hyzaar^®^ tablets as a control. Following oral administration in beagle dogs, there were no significant differences in bioavailability between the No. 1 tablet and the Hyzaar^®^ tablet in HCTZ, losartan carboxylic acid (E-3174), and LOS-K (F < F_0.05_). Thus, formulation and preparation factors were determined according to the combination of the compression and mechanical properties of the mixed powder and quality of tablets, which was demonstrated to be a feasible method in direct powder compression.

## 1. Introduction

The hydroxychlorothiazide (HCTZ)/losartan potassium (LOS-K) tablet was the first compound preparation comprised of an antagonist to angiotensin II receptor (AT1) and a diuretic, which was approved by the FDA in April 1995. The combination of HCTZ and LOS-K exhibited a synergistic effect on pharmacological activity, increased antihypertensive efficacy, and reduced adverse reactions [1,2].

HCTZ is a class-IV drug according to the Biopharmaceutics Classification System, having low solubility and low permeability, and, consequently, low absorption in the gastrointestinal tract [3]. To improve the dissolution rate and bioavailability of HCTZ, inclusion complexes [4,5], co-precipitate systems [6], solid–lipid nanoparticles [7], isomalt-based moulded tablets [3], and freeze-dried rapidly-disintegrating tablets/lyophilized emulsion tablets [8] have been developed and have exhibited good performance. In addition, a decrease in particle size and an increase in surface area are efficient technologies to improve the dissolution rate and bioavailability of poorly water-soluble drugs [9].

The commonly used HCTZ/LOS-K formulation is a tablet prepared by processing technology made of wet granulation or dry granulation. However, HCTZ is sensitive to moisture and is easily hydrolysed by moisture or heat due to its sulfonamide [10]. Thus, wet granulation is not the optimal choice for HCTZ. It is speculated that direct powder compression is suitable for HCTZ/LOS-K tablets, with desirable stability during production and long-term storage.

Direct powder compression is a well-known and simple method in tablet manufacturing. The production process includes mixing and pressing [11]. There is growing evidence that direct powder compression has become the first choice for oral solid preparations due to its short development cycle and low cost. A series of excipients equipped with excellent compressibility and flowability have been developed for direct powder compression. The compression and mechanical properties of mixed powders are key factors for the process of direct compression. There might be some differences in the drug dissolution rate and bioavailability between tablets prepared in mass production and testing. This may be caused by the different porosities and tensile strengths of tablets and the deformation mechanism of excipients/drugs, which are important but easily neglected.

To produce tablets using direct compression, it is imperative to understand the compression and mechanical properties of the mixed powder (drug and excipient), including their compressibility and compactibility. Compressibility is the ability of powder to deform or decrease in volume under pressure, while compactibility refers to the powder’s ability to be compressed into a compact with some mechanical strength [12,13]. A variety of methods and parameters are used to assess the mechanical properties of powders, such as the stress-relaxation curve, Kawakita equation, Shapiro equation, Heckel analysis, and yield pressure (Py). Many studies have investigated the mechanical properties of a single excipient or drug to guide formulation screening in direct powder compression. The compression behaviour classification system for pharmaceutical powders was successfully built, and a series of standardized approaches for powder compression analysis were established [14]. The compressibility and compactibility of mixed powder were inconsistent in different studies due to their complexity [15,16].

In this study, HCTZ/LOS-K tablets with improved bioavailability of HCTZ were prepared by direct powder compression. To improve the feasibility of direct powder compression, the type and number of excipients are screened by testing the compression and mechanical properties of the mixed powders. The mechanical properties and quality of the tablets were used to determine the compaction pressure and tableting speed. The particle size of HCTZ was investigated in terms of the compressibility and HCTZ dissolution rate. According to the results of the dissolution rate in nine formulations, a D-optimal mixture experimental design was used to select the optimal formulation. Furthermore, the in vitro dissolution curve and relative bioavailability in vivo of the optimal formulation were compared with Hyzaar^®^ tablets (MSD R & D (Shanghai, China) Co., Ltd.) to confirm the quality of the tablets in mass production. The compression and mechanical properties were basic and important characters for mixed powder, which should be given more attention in the preparation of oral drugs. The formulation and preparation factors were determined according to the combination of the compression and mechanical properties of mixed powder and quality of tablets, which might be a feasible method in direct powder compression using other mixed powders.

## 2. Materials and Methods

### 2.1. Materials

LOS-K and HCTZ were provided by Zhejiang Tianyu Pharmaceutical Co., Ltd. (Taizhou, China). Microcrystalline cellulose (MCC PH102) was provided by Asahi Kasei Pharmaceutical Co., Ltd. (Tokyo, Japan). Lactose monohydrate (SuperTab^®^14SD) was sourced from Dfe Pharma India Pvt.ltd. (Tamil Nadu, India). Pre-gelatinized starch (Starch 1500 ^®^) and Opadry^®^ were provided by Colorcon.

### 2.2. Mechanical Properties of the Mixed Powder

Tablets (250–260 mg) were compressed using a single-punch tablet press (Romaco, Styl’one Evolution, Karlsruhe, Germany) with a flat punch, EuroB-11.28 (99.93 mm^2^).

Stress relaxation curve: The stress relaxation curve was simulated by a powder compression characteristic tester (MPC100, OKADA SEIKO, Tokyo, Japan). Forty milligrams of powder were put into the powder compression characteristic tester for 4 min. The diameter of the punch die was 5 mm, and the speed of the upper punch was set as 0.1 mm/min. In the stress-relaxation curve, the horizontal axis shows the time and the vertical axis shows the pressure of the upper punch. The formulas for the stress-relaxation test are as follows [17]:Y_(t)_ = (F_(0)_ − F_(t)_)/F_(0)_(1)
t/Y_(t)_ = 1/(ab) + (1/a) × t(2)
where Y_(t)_ is a parameter of stress relaxation, F_(t)_ is the maximum pressure, t is the time from pressure reaching the maximum, and a and b are constants.

The Kawakita equation: The Kawakita equation was developed to describe the relationship between the volume reduction of the powder and the applied pressure to characterize the powder compressibility [18,19]: C = (V_0_ − V)/V_0_(3)
P/C = 1/(ab) + P/a(4)
where V_0_ is the initial volume of the powder bed, V is the powder volume after the application of compression pressure (P, 25–250 MPa), C is the volume reduction degree under P, and a and b are constants that are related to the compactibility and resistant forces to P, respectively.

Shapiro equation: As a model of the powder compression process, the Shapiro compression parameter f is derived from the Shapiro equation [20]:Ln (E) = Ln (E_0_) – kP − fP^0.5^(5)
E = 1 − ρ_A_/ρ_T_(6)
where E is the porosity of the powder bed, E_0_ is the initial porosity of the powder bed, P is the applied compression pressure (0–50 MPa), and k and f are constants. Furthermore, ρ_A_ is the compact density and ρ_T_ is the true density of the material.

Heckel analysis: [21] Heckel analysis is among the most popular methods to characterize the compressibility behaviour of blends during compression:In 1/(1 − D) = kP + A(7)
D = ρ_A_/ρ_T_(8)
where A is a constant, P is the applied compression pressure, K is the slope of the linear part of the plot and is inversely related to the Py or yield stress, D is the relative compact density (solid fraction) at compression pressure P, ρ_A_ is the compact density, and ρ_T_ is the true density of the material. The value of 1-D is the porosity of the powder bed.

Calculation of the energy during the process of pressing: 40 mg of powder was put into the powder compression characteristic tester (MPC100, O KADA SEIKO, Japan). The diameter of the punch die was 5 mm and the speed of the upper punch was set as 0.1 mm/s. The software, Analis (Medelpharm, Beynost, France), was used to calculate the energy.

### 2.3. Characteristics of Tablets

Elastic recovery (ER): The axial elastic recovery percentage E_R_ (in-die) is given by:E_R_ = (H − H_0_)/H_0_(9)
where H_0_ is the tablet thickness at the minimal punch distance and H is the truly measured table thickness after ejection.

#### 2.3.1. Tensile Strength (Ts)

T_s_ = 2F/(π · D · L)(10)
where F is the tablet crushing force (hardness), and the Romaco tablet press records diametrical crushing force (F), tablet diameter (D), and thickness (L).

#### 2.3.2. Tablet Porosity (ε)

The porosity of a tablet can be calculated during compaction (in-die) and after compaction (out-of-die):ε_(in die)_ = 1 − ρ_in die_/ρ_true density_(11)
ε_(out die)_ = 1 − ρ_out die_/ρ_true density_(12)
where ρ_in die_ is the density of tablets in-die, ρ_out die_ is the density of tablets out-of-die, and ρ_true density_ is the true density of the blends.

#### 2.3.3. Fragility Test

Powders on the surface of tablets were removed. Then, these tablets were accurately weighed (W_0_, 6.5 g) and transferred into the fragility tester (25 rpm, 4 min). After that, the floating powders were removed, and these tablets were accurately weighed (W) again.
(13)$$\mathrm{Friability}=\frac{W_{0}-W}{W_{0}}\times 100\%$$

#### 2.3.4. Dissolution Rate

Distilled water with pH 1.2/pH 4.5/pH 6.8 solution (900 mL, 37 °C) was used as the medium in this study. In pH 1.2 medium solution, 3 mL of medium was removed from the dissolution apparatus and immediately replaced with 3 mL of fresh medium at 15, 30, 45, 60, 90, and 120 min. In water/pH 4.5/pH 6.8 medium solution, samples were removed at 5, 10, 15, 20, 25, 30, and 60 min. Each sample was filtered and analysed by high-pressure liquid chromatography.

### 2.4. D-Optimal Mixture Experimental Design

The effect of lactose monohydrate (SuperTab^®^14SD, wt.%, A), MCC (PH102, wt.%, B), and pre-gelatinized starch (Starch-1500, wt.%, C) on the drug dissolution rate was studied by D-optimal mixture experimental design (Design Expert software, version 7.0, Stat-Ease, Inc., Minneapolis, MN, USA). According to our previous results of reverse engineering and the literature, the contents of LOS-K are fixed at 20%, and HCTZ is 5% and magnesium stearate is 0.7%. Furthermore, the total content of lactose monohydrate, MCC, and pre-gelatinized starch is 74.3%. The design matrix was generated based on the upper and lower limits for A (18.11–32.4 %), B (27.23–42.92%), and C (10.02–17.92%). A total of nine experimental runs were designed, as shown in Table 1. The dissolution rate of the R1–R9 preparation was recorded and used for mathematical model imitation. According to the mathematical models, further formulation optimization was performed by Design-Expert^®^ 7.0. A, B, and C were used as independent variables and the dissolution rate was used as a dependent variable. The mathematical models were used to imitate the drug dissolution by Design Expert ^®^ 7.0.

### 2.5. Pharmacokinetics and Bioavailability Study

Six beagle dogs (males, 8.73 ± 0.92 kg) were randomly divided into two groups and given Hyzaar^®^ (as a reference) and No. 1 tablet (LOS-K: 50 mg per tablet, HCTZ: 12.5 mg per tablet). A single-dose, double-period, randomized crossover design method was established, and the washout period was one week. Both groups received oral administration of Hyzaar^®^ and No. 1 tablets with 60 mL of water after a 12-h fast. Blood samples (3.0 mL) were collected from a foreleg vein and dropped into heparinized centrifuge tubes at 0.33, 0.67, 1.0, 1.33, 1.67, 2.0, 2.5, 3.0, 3.5, 4.0, 5.0, 6.0, 8.0, 12.0, 24.0, and 36.0 h following administration. Blood samples were centrifuged at 4000 rpm/min for approximately 10 min. The plasma was removed and stored at −20 °C until analysis. Losartan carboxylic acid (E-3174) is an active metabolite of potassium losartan by the CYP2C9 enzyme, and the activity of E-3174 is approximately 10–40 times higher than that of potassium losartan. The detection of plasma samples (potassium losartan, E-3174 and HCTZ) was performed using an ACQUITYTM UPLC system (Waters Corp., Milford, MA, USA) after the liquid–liquid extraction procedure.

### 2.6. Statistical Analysis

All data were analysed and expressed as the mean ± standard deviation (SD). The pharmacokinetic data were analysed by drug and statistics (DAS) version software.

## 3. Results and Discussion

The compression and mechanical properties were basic and important characters for mixed powder, which should be given more attention in the preparation of oral drugs. 

### 3.1. Selection of Lactose Monohydrates and Magnesium Stearate

#### 3.1.1. Lactose Monohydrates Exhibited Different Stress–Relaxation Curves

In solid dosage forms, lactose is probably the oldest but still one of the most important diluents in tableting [22]. In addition to good flow properties and compressibility, lactose also has low sensitivity to lubricants and strain rate. Lactose monohydrates (FlowLac^®^100), obtained by spray drying technology, show better flow properties, larger surface areas, and improved compressibility. Lactose monohydrates (SuperTab^®^30GR) consist of α-lactose and β-lactose. A certain amount of β-lactose enhances the compressibility of SuperTab^®^30GR, and it’s stability and hygroscopicity are better due to the absence of amorphous lactose crystals. Lactose monohydrates (SuperTab^®^14SD) are obtained by spray drying a suspension of lactose monohydrates with excellent flow properties and compressibility.

The tablet was pressed up to a certain compacting pressure, and after reaching this pressure, the punches were stopped for some time period. At this moment, the decrease in compacting pressure was measured, which was called stress relaxation [23,24,25]. The greater the degree of stress relaxation indicated that more energy was required for plastic deformation and reduced the risk of capping. The stress-relaxation curves of FlowLac^®^100, SuperTab^®^30GR, and SuperTab^®^14SD were fitted using the second-order exponential decay function (Figure 1). Generally, powders with stronger plastic deformation exhibit a more obvious decay process.

The stress-relaxation curves could be divided into two stages: the compression stage (I) and stress-relaxation stage (II). During the compression stage, powders were pressed by the upper punch. The plastic property is reflected by the time spent reaching the highest pressure, and a longer time represents a weaker plastic property. From Figure 1A, the three types of lactose monohydrate spent almost the same amount of time at the compression stage. During the stress-relaxation stage, more rapid decay reflected better plastic deformation. From Figure 1B, the order of plastic deformation in different types of lactose monohydrate is SuperTab^®^14SD > SuperTab^®^30GR > FlowLac^®^100.

#### 3.1.2. Effect of Magnesium Stearate on Compressibility of the Mixed Powder

Magnesium stearate has been widely used as a lubricant in solid formulations. MCC and pre-gelatinized starch are typical excipients sensitive to lubricant [26]. Thus, the amounts of magnesium stearate and mixing time have a significant influence on the tensile strength of the mixed powder. According to Section 2.4, R7 powder (without adding magnesium stearate) was used for the study of magnesium stearate. The effects of the magnesium stearate amount and mixing time on the tensile strength and Py value of the mixed powder are shown in Figure 2.

The Py value increased with mixing time. When the mixing time was longer than 1 min, the Py value increased significantly, and both compressibility and compactibility decreased. With the increase in the magnesium stearate amount, the Py value did not change obviously. However, the tensile strength decreased significantly when the amount was higher than 0.7%.

### 3.2. Compressibility of the Mixed Powder in Different Proportions

#### 3.2.1. Stress-Relaxation Curves

The stress-relaxation curves of the R1–R9 mixed powder (shown in Figure 3 were fitted by MPC-100, as described in Section 2.2.

The stress-relaxation formulas of R1–R9 are shown in Table 2. A larger a-value represents better stress relaxation and compressibility. Thus, the order of stress relaxation and compressibility is R8 > R6 > R7 > R9 > R4 > R5 > R2 > R3 > R1. The a-value was fitted with the contents of MCC PH102, lactose monohydrate (SuperTab^®^14SD), and pre-gelatinized starch. The fitted results are shown in Figure 4A.

Stress relaxation increased with the total proportion of MCC. The fitted formula is a = 0.1226x + 0.1523y + 0.1853z − 0.09412xz − 0.14yz + 0.31172xyz. The contribution rate of each factor to stress relaxation was determined as follows: pre-gelatinized starch > MCC > lactose monohydrate.

#### 3.2.2. Py Value

The Py values of R1–R9 are shown in Table 3. The fitted formula is Py = 11.69x + 4.23y + 29.29z − 0.23xy − 0.80xz − 0.49xyz. The contribution rate of each factor to stress relaxation was in the order pre-gelatinized starch > lactose monohydrate > MCC (Figure 4B). Thus, the powder showed preferable compressibility when the content of pre-gelatinized starch was 10.96–15.91%, MCC was 32.86–43.31%, and lactose monohydrate was 19.53–26.91%.

Both stress-relaxation curves and yield stress were used to evaluate the compressibility of R1–R9. However, the results were not completely consistent. This may be because the speed of the upper punch (0.1 mm/min) for the stress-relaxation curves was slow. R7 was selected and used in the following research of the compression mechanism.

### 3.3. Compression Mechanism of the Mixed Powder

#### 3.3.1. Prediction of Rearrangement Tendency by the Kawakita Equation

The compression mechanism is determined by the type of powder, e.g., the deformation mechanism of plastic materials (MCC) is plastic deformation, and for brittle materials (lactose), it is fragmentation. Confirming the type of mixed powder will help predict their deformation mechanism.

The rearrangement tendency of the mixed powder under 25–250 MPa was evaluated by determining the a- and b-values in the Kawakita equation (P/C = 1.38969 × P + 12.8402). In the compressibility index, a represents the maximum degree of volume reduction [13,14,27,28,29]. Parameter b^−1^, representing cohesiveness or plasticity, is the pressure when the value of C reaches 1/2 of the limiting value. A higher 1/b value represents more plasticity or less resistance to compression. It has been demonstrated that a large ab-value results in a high degree of particle rearrangement. From Figure 5A, the rearrangement index ab was 0.078 < 0.1, meaning that the mixed powder showed nearly no initial particle rearrangement. Thus, the mixed powder can be classified as a Class-II powder [30,31].

#### 3.3.2. Prediction of Fragmentation Tendency by the Shapiro Equation

The Shapiro equation (Ln(E) = −0.3346−0.00268 × P + 0.15126 × P^0.5^) and Shapiro parameter f (f = 0.15126 > 0.1) were obtained from Figure 5B when the compression pressures were in the range of 0–50 MPa. For a Class-II powder, the parameter f is used to indicate particle fragmentation that occurs during compression [14,27]. According to parameter f, Class-II powders are classified as Type-IIA (f < 0.1, ductile, low degree of fragmentation) or -IIB (f > 0.1, brittle, high degree of fragmentation). Thus, it is speculated that the compression mechanism of the mixed powder is as follows: under low pressure, particle fragmentation and secondary rearrangement of fragmented particles dominate.

Lactose monohydrate (FlowLac^®^100), starch 1500^®^, and MCC PH102 are classified as Class-IIB powders, Class-IIA powders, and Class-I powders, respectively. Thus, both mechanical properties and the reduction in volume of the mixed powder in the low-pressure region are mainly affected by FlowLac^®^100.

#### 3.3.3. Heckel Analysis and the Value of Apparent Mean Py

To assess the tableting performance of the mixed powder, a Heckel analysis was carried out, and the value of Py was determined. From Figure 5C, region II is in the range of 96–296 MPa. Py, the reciprocal of the slope of the Heckel equation, is an indicator of the compressibility of the material. The mean value of Py is 139.4 MPa, which indicated that the mixed powder was moderately hard. Thus, it belongs to Type-II powders according to the Heckel curve. In Region I (0.5–96 MPa), the particle fragmentation and arrangement of fragmented particles dominate. According to the Py values of excipients (FlowLac^®^100:168 MPa, Starch 1500^®^: 67.9 MPa, MCC PH102: 91.3 MPa) [32], it is speculated that the compressibility of the mixed powder is determined by FlowLac^®^100.

#### 3.3.4. Force-Displacement Curve and Compressive Work

A force-displacement curve (shown in Figure 6) was used for calculating the work during the pressing process. The changes in compressive work, elastic work, and plastic work at different pressures are shown in Table 4. When the pressure is in the range of 4.33–30.73 kN, both plastic work and elastic work increase with the pressure. Compressive work is mainly affected by plastic work. After the pressure reaches 37.29 kN, the elastic work increases continuously, and the plastic work remains the same. It is speculated that elastic deformation occupies the main parts and that a lot of energy is consumed when the pressure is higher than 37.29 kN. Thus, high pressure is not beneficial to the formation of tablets, which is consistent with the predicted result from the Heckel curve.

### 3.4. Effect of Compaction Pressure

The curves of mean stress-tensile strength, mean stress-elastic recovery, and mean stress-porosity (in-die and out-of-die) are shown in Figure 7. An increase in the pressure affected the interaction between powders and the hardness and tensile strength of tablets. Generally, the elastic recovery of tablets should be between 2–10%. If the elastic recovery is higher than 10%, over-compression may result in a reduction in the tensile strength. From Figure 7A, in the range of 50–150 MPa, elastic recovery decreased with compaction pressure. This may be because the fragmentation of particles and increase in contact area led to the enhanced combination and interaction between powders. From 150 to 300 MPa, the elastic recovery did not change in the process of plastic deformation. The elastic recovery increased obviously in the process of elastic deformation at high compaction pressures (>300 MPa).

The adhesion between powders can be predicted by the tensile strength values. Generally, a high tensile strength value indicates a tight bond between powders. The desirable value of tensile strength is higher than 1–2 MPa [33]. The tensile strength value increased with the degree of compaction pressure until 300 MPa. However, it did not increase at high compaction pressures (>300 MPa) due to the stable binding capacity of particles in tablets. High compaction pressures might lead to a slow drug dissolution rate from ibuprofen tablets prepared by roller compaction and tableting [34].

Porosity not only can indicate the compressibility of materials, but also affects the shelf-life stability and dissolution rate of the compressed tablets. From Figure 7B, porosity both in-die and out-of-die decreased with the degree of compaction pressure. The difference between the porosity in-die and out-of-die has the same tendency for change with elastic recovery. The porosity of tablets affects the rate of medium solution entering into the tablets and, consequently, their dissolution curve.

### 3.5. Effect of Tableting Speed

The effects of the tableting speed were tested to confirm the deformation mechanism of the mixed powder and ensure the quality of the tablets. Tablets (250–260 mg) were compressed using a Romaco tablet press with a flat punch, EuroB-11.28. The mean pressure was set as 20 kN, and the tableting speed rate was 600–4800 tabs/h. The real compression time consisted of compression time (pressure: 0-maximum) and decompression time (pressure: maximum-0), as shown in Table 5.

The effect of the tableting speed on plastic materials is greater than that on brittle materials because plastic deformation is time-dependent. From Figure 8 The Py value was in the range of 128–141 MPa, which was not seriously affected by the tableting speed. It was confirmed that the mixed powder tends to be brittle, which was consistent with the result of the Shapiro compression equation.

### 3.6. Effect of Particle Size on HCTZ Dissolution Rate and Properties of the Mixed Powder

To improve the dissolution rate of HCTZ (the solubility is approximately 722 mg/mL [35]), a jet mill was used to prepare drug particles at the nanoscale. In a jet mill, a high-velocity air stream results in acute collisions between particles. A narrow particle size distribution can be obtained after particles leave the mill as soon as their size is reduced to the size of a classifying exit. Particles of different sizes were prepared due to different Venturi and Ring pressures. A Romaco tablet press (mean pressure: 30 kN) was used to investigate the effect of the particle size of HCTZ on the properties of the mixed powder. The particle size distribution of HCTZ and dissolution rate of tablets prepared are shown in Table 6. The particle size decreased with the Venturi and Ring pressure, and consequently, the drug dissolution rate increased. On the other hand, with the decrease in the particle size of HCTZ, the ab-value and the rearrangement tendency of the mixed powder increased. All mixed powders containing HCTZ with different particle sizes showed f values larger than 0.1, which indicated that the mechanism of deformation was not affected by particle size. The Py value increased with the particle size. This may be explained by the fact that for small particles, fragmentation decreases and rigidity increases. The compressibility of small particles decreased. Thus, the Venturi and Ring pressures were set in the range of 3–3.5 bar and 2–2.5 bar, respectively, to ensure that the mixed powder had good compressibility.

### 3.7. Dissolution Rate of the Optimal Tablets Selected by the D-optimal Mixture Experimental Design

R1–R9 tablets (Table 1) were prepared according to the technological parameters selected above. A D-optimal mixture experimental design was used to select an optimal formulation based on the dissolution results of R1–R9 tablets (not shown). The D-optimal mixture experimental design utilized mathematical and statistical methods to optimize the formulation. Multiple variables are simultaneously studied through a minimum number of observations, time, and cost. Thus, a D-optimal mixture experimental design is widely used to investigate the interaction between the independent and response variables in the optimized formulation [36,37,38]. The dependent variables (response) were the cumulative dissolution of HCTZ/LOS-K in water at 20 min and pH 1.2 medium solution at 120 min. The target values were set as the cumulative dissolution of Hyzaar^®^ (water, at 20 min) and the maximum (pH 1.2 medium solution, at 120 min), respectively. An optimal formulation (No. 1) was obtained: 26.027% (wt.%) lactose monohydrate, 32.811% (wt.%) MCC PH102, and 15.462% (wt.%) pre-gelatinized starch. No. 1 tablets (mass production, 10,000 tablets) were produced by direct compression using a rotary tablet press (PT-3P380V5OBz, Liaoning Tianyi Machinery Co., Ltd. Shenyang, China). According to the selected technique parameters above, the compression pressures were 20–25 kN, and the tableting speed was 19.5–22.5 kT/h. In addition, to inhibit the moisture absorption and increase the stability of tablets, No. 1 tablets were coated by Opadry with 2.78% coating weight. The mean weight of the obtained No. 1 tablet (mass production) was 249.68 ± 5 mg, with appropriate fragility (0.14%), hardness (95 ± 6 N), thickness (4.32 ± 0.4 mm), and moisture content (3.2%). The dissolution curves of the No. 1 tablet (mass production) and Hyzaar^®^ are shown in Figure 9. In pH 1.2 medium solution, LOS-K dissociates and losartan-free acid precipitates are used as a gel layer around the tablet, which results in insufficient tablets in other media. For HCTZ, the dissolution rate from No. 1 tablets (mass production, 10,000 tablets) in all media was faster, and the cumulative release at the endpoint was greater than that from Hyzaar^®^. f_2_ factors of HCTZ from No. 1 tablets (mass production) and Hyzaar^®^ were 75, 68, 68, and 64, in pH 1.2, 4.5, and 6.8 medium solution and water, respectively. f_2_ factors of LOS-K from No. 1 tablets (mass production) and Hyzaar^®^ were 95, 62, 72, and 79, in pH 1.2, 4.5, and 6.8 medium solution and water, respectively.

### 3.8. Pharmacokinetic Study

A single-dose, double-period, randomized crossover method was used to investigate the pharmacokinetic curves of the No. 1 tablet (mass production) and Hyzaar^®^ (Figure 10). Compared with Hyzaar^®^, the relative bioavailability of the No. 1 tablet (mass production) was 103.85% (LOS-K), 88.94% (E-3174), and 104.29% (HCTZ). The T_max_ of HCTZ (1.332 ± 0.36 h) and LOS-K (1.167 ± 0.28 h) in the No. 1 tablet (mass production) was later than that in the Hyzaar^®^ tablet (HCTZ, 1.277 ± 0.25 h, LOS-K, 1.000 ± 0.36 h), and the C_max_ of HCTZ (1017.50 ± 193.89 ng/mL) and LOS-K (487.45 ± 235.69 ng/mL) in the No. 1 tablet (mass production) was lower than that in the Hyzaar^®^ tablet (HCTZ, 1080.17 ± 187.54 ng/mL, LOS-K, 598.92 ± 410.39 ng/mL). It is speculated that the lower C_max_ and delayed T_max_ can be adjusted by increasing the amounts of lactose monohydrate and pre-gelatinized starch in the formulation.

The pharmacokinetic parameters were tested for significance by multifactor analysis of variance (ANOVA) after logarithmic transformation. Then, the bioavailability of the No. 1 tablet (mass production) was determined by two one-sided t-tests and analysis of the 90%-confidence interval. Following oral administration, there was no significant difference between No. 1 tablets (mass production) and Hyzaar^®^ in HCTZ, E-3174, and LOS-K (F < F_0.05_). It was consistent with the results of drug dissolution rates, and there was no significant difference between the No. 1 tablet and Hyzaar^®^ tablet. For LOS-K and E-3174, the individual differences were significant due to their highly variable coefficients (35.74%). In the following research, it is necessary to calculate the variable coefficients of Hyzaar^®^ and eliminate individual differences to make the test data more accurate. The formulation and preparation factors were determined according to the combination of the compression and mechanical properties of mixed powder and quality of tablets, which might be a feasible method in direct powder compression using other mixed powders.

More and more attention has been paid to establish large datasets of pharmaceutical materials with compression and mechanical properties [39,40,41]. Thoorens et al. described a database of 84 MCC materials with 10 physicochemical properties [42]. Hayashi et al. established a dataset containing 81 kinds of active pharmaceutical ingredient (API) characteristics and obtained a non-linear regression model fitted by the boosted-tree algorithm with satisfied calibration and prediction performance [43]. In 2019, Dai et al. built the compression behavior classification system for pharmaceutical powders and established a series of standardized approaches for powder compression analysis [14]. Instead of building large datasets of pharmaceutical materials, in this paper a specific formulation was obtained according to the combination of the compression and mechanical properties of mixed powder and the quality of tablet. It was an improvement that guided drug production utilizing basic theoretical knowledge (compression and mechanical properties of mixed powders).

## 4. Conclusions

The compression and mechanical properties of the mixed powder were important for the feasibility of direct comparison and the qualities of the tablets. In this study, the compression and mechanical properties of the mixed powder and the quality of tablets were investigated to determine the formulation and technology of direct compression, on the basis of which a D-optimal mixture experimental design was used to further select the optimal formulation. The optimal No. 1 tablet (mass production) exhibited a desirable drug dissolution rate and bioavailability compared with Hyzaar^®^ tablets. Thus, it was demonstrated that formulation and technology were determined according to the combination of the compression and mechanical properties of mixed powder and the quality of tablets, which was a feasible method in direct powder compression.

## Figures and Tables

**Figure 1 pharmaceutics-14-01741-f001:**
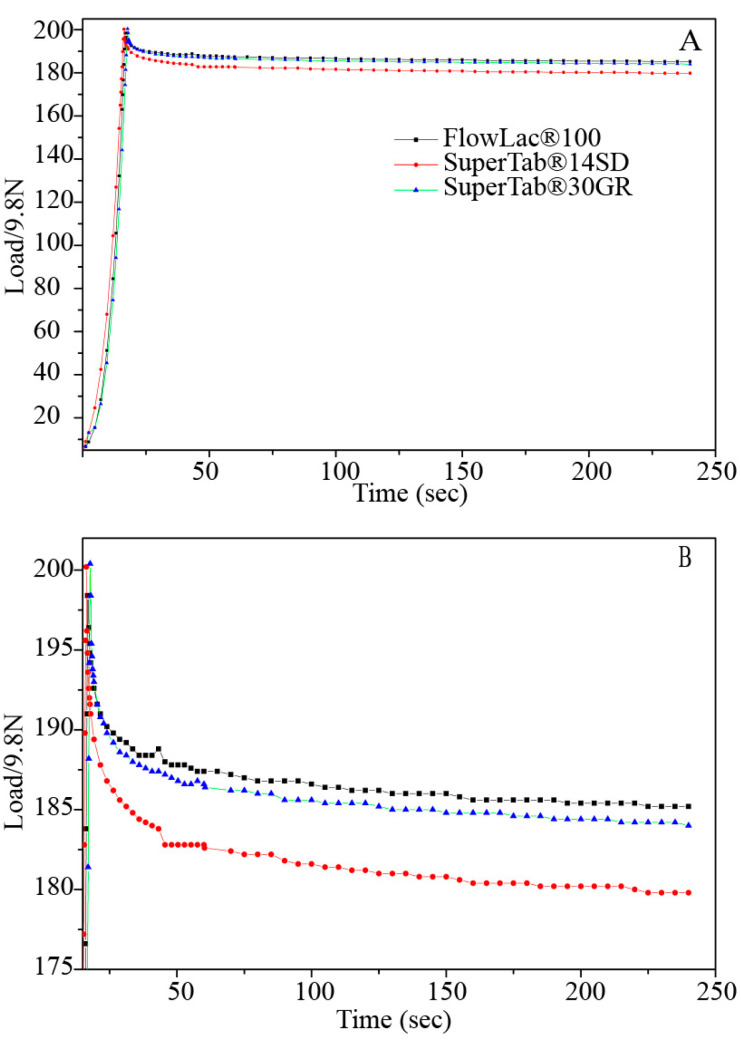
(**A**) The stress-relaxation curves of lactose monohydrate (Y-axis: 0–200 s). (**B**) The stress-relaxation curves of lactose monohydrate (Y-axis: 175–200 s).

**Figure 2 pharmaceutics-14-01741-f002:**
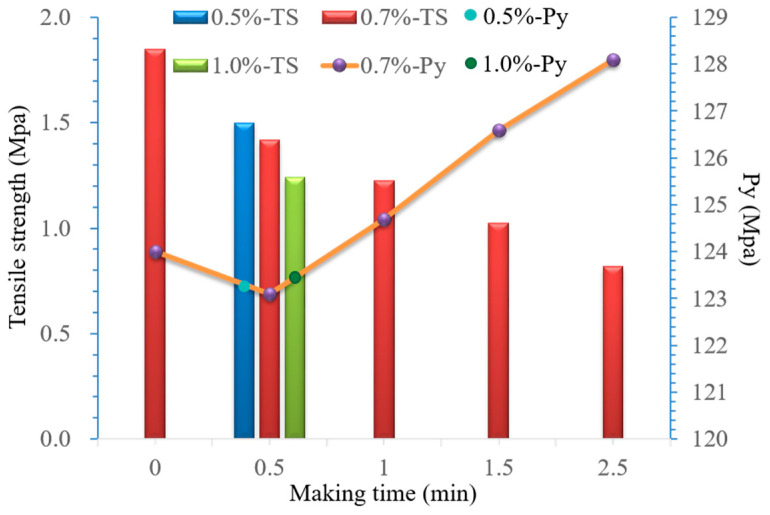
Effect of amount (0.50%, 0.70%, 1%) and mixing time of magnesium stearate on tensile strength and Py value.

**Figure 3 pharmaceutics-14-01741-f003:**
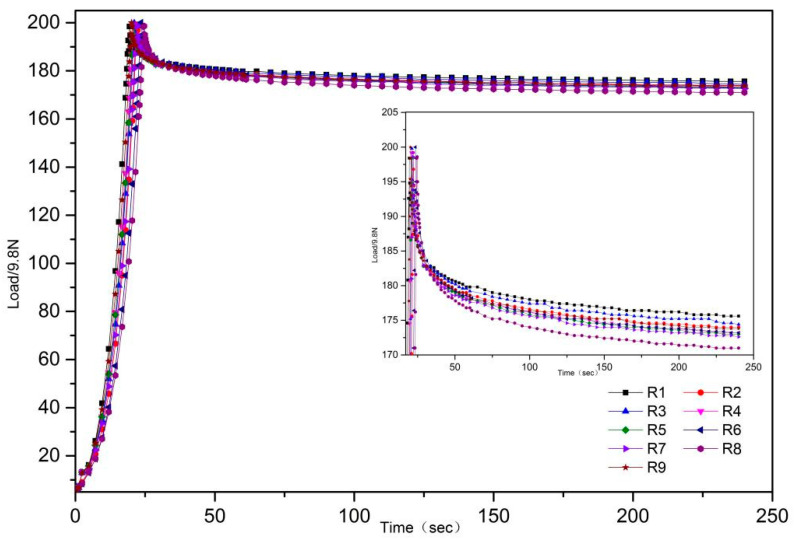
The stress-relaxation curves of R1–R9.

**Figure 4 pharmaceutics-14-01741-f004:**
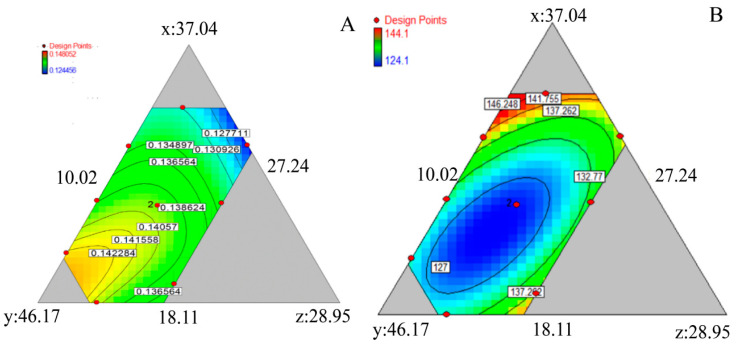
Effects of each factors on a value (**A**) and Py value (**B**). x: lactose monohydrate (SuperTab^®^14SD), y: MCC PH102, z: pre-gelatinized starch.

**Figure 5 pharmaceutics-14-01741-f005:**
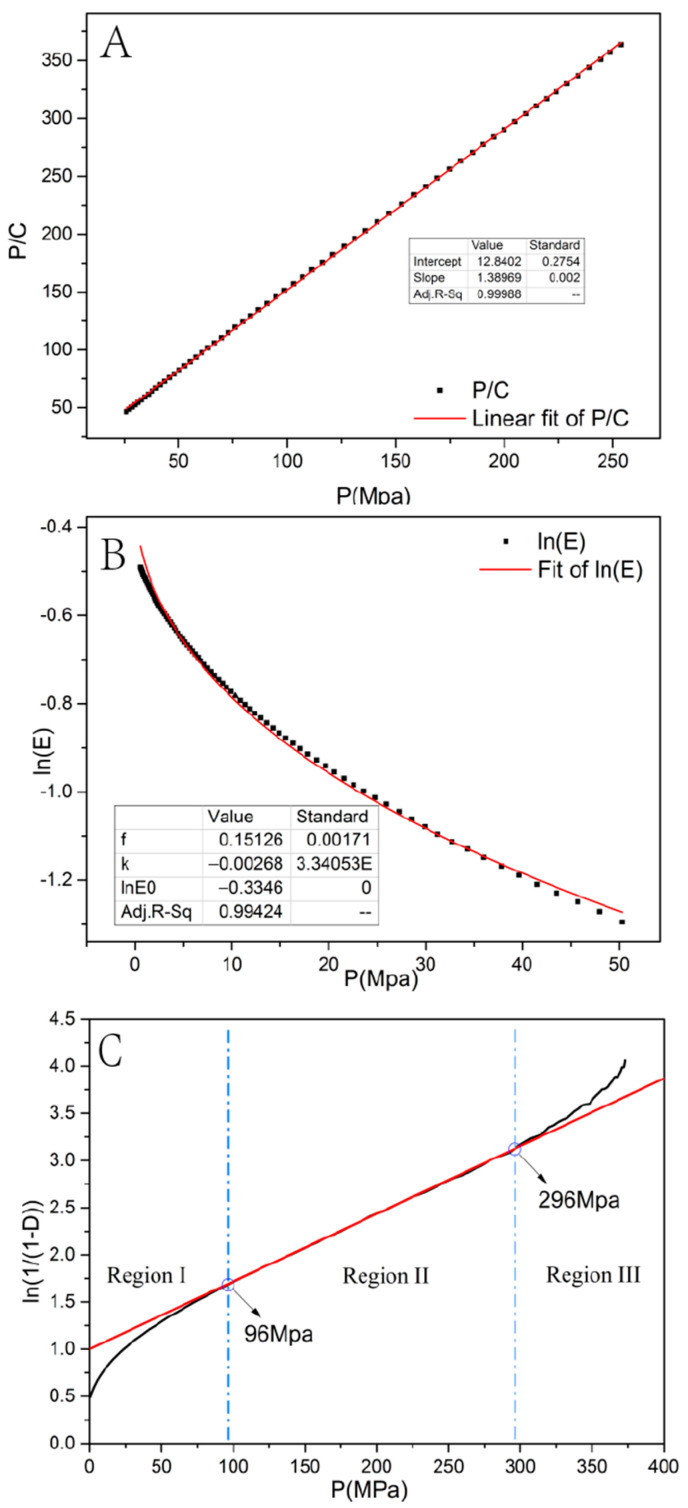
The fitting curve of Kawakita equation (**A**), Shapiro equation (**B**), and Heckle profile (**C**) of the mixed powder.

**Figure 6 pharmaceutics-14-01741-f006:**
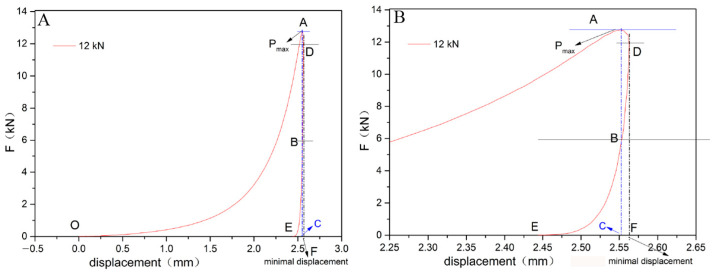
(**A**) The force-displacement curve of the mixed powder under compression pressure of 12 kN. (**B**) Detail in enlarged scale (2.25–2.65 mm).

**Figure 7 pharmaceutics-14-01741-f007:**
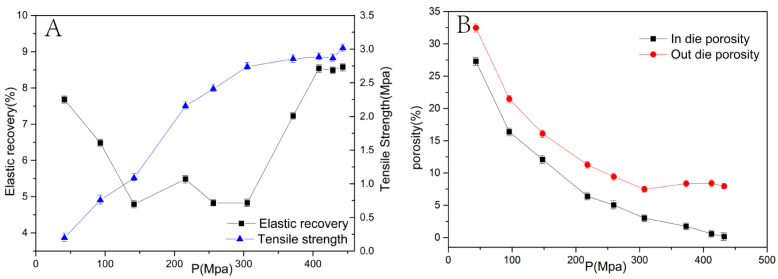
The effect of compaction pressure on elastic recovery, tensile strength (**A**), and porosity (**B**).

**Figure 8 pharmaceutics-14-01741-f008:**
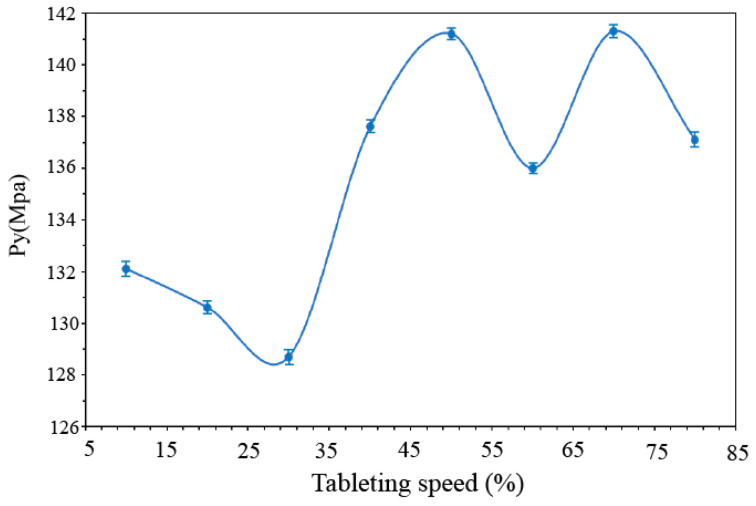
The effect of tableting speed on Py value.

**Figure 9 pharmaceutics-14-01741-f009:**
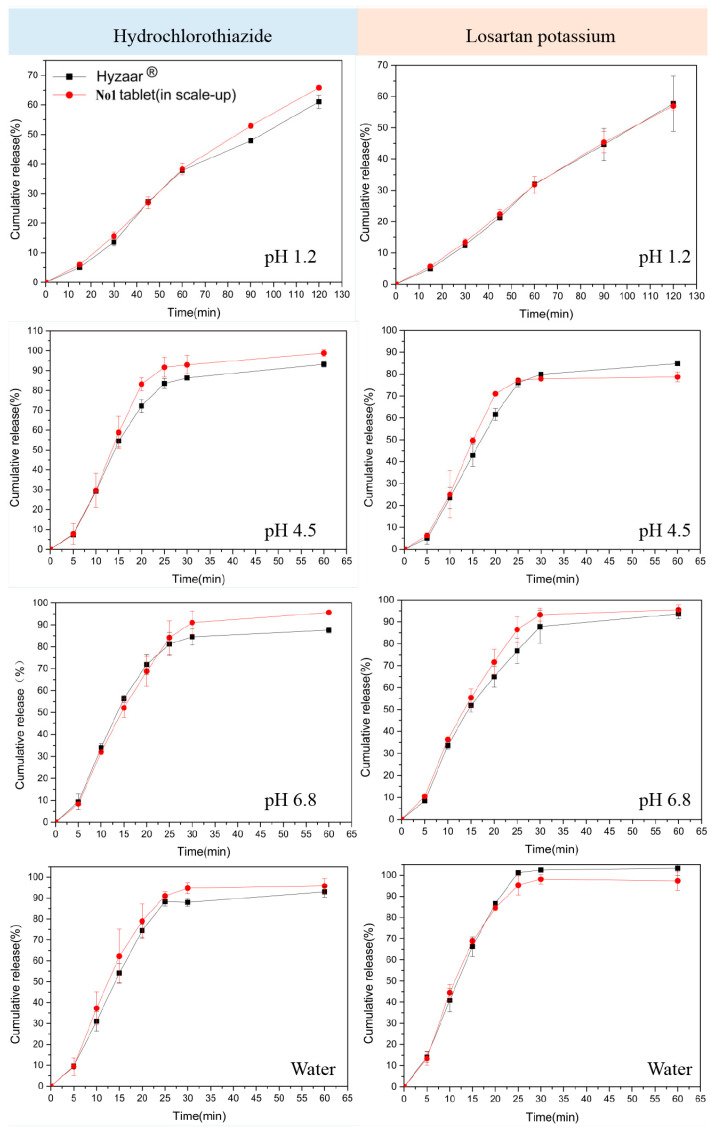
HCTZ/LOS-K dissolution rate in different medium solution.

**Figure 10 pharmaceutics-14-01741-f010:**
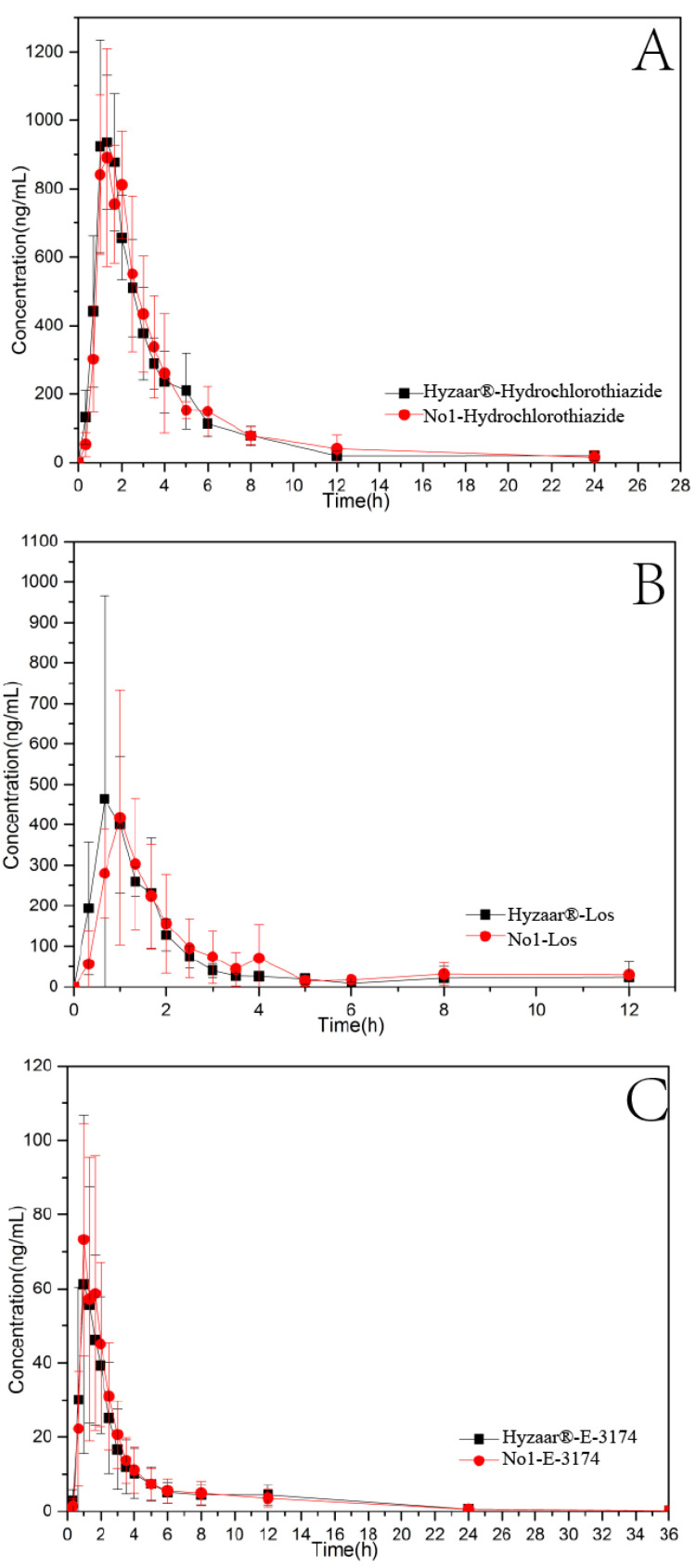
Mean plasma-concentration time profiles of HCTZ (**A**), LOS-K (**B**), and E-3174 (**C**) after oral administration.

**Table 1 pharmaceutics-14-01741-t001:** Design for mixture experiment.

Factor/Run	Low (%)	High (%)	R1	R2	R3	R4	R5	R6	R7	R8	R9
A (%)	18.11	32.4	29.646	29.572	32.4	19.47	18.11	25.583	25.052	21.745	25.402
B (%)	27.23	42.92	27.24	34.708	29.907	36.91	42.444	38.697	35.348	42.535	30.978
C (%)	10.02	17.92	17.414	10.02	11.993	17.92	13.746	10.02	13.9	10.02	17.92

**Table 2 pharmaceutics-14-01741-t002:** Stress-relaxation equation in R1–R9.

	Equation	R^2^	a
R1	y = 8.0350x + 55.10	0.9985	0.1245
R2	y = 7.5295x + 49.64	0.9986	0.1328
R3	y = 7.7512x + 53.74	0.9983	0.1290
R4	y = 7.3073x + 53.39	0.9984	0.1368
R5	y = 7.3508x + 53.85	0.9984	0.1360
R6	y = 7.1440x + 48.77	0.9987	0.1400
R7	y = 7.2264x + 50.53	0.9985	0.1384
R8	y = 6.7544x + 45.47	0.9986	0.1481
R9	y = 7.272x + 50.11	0.9987	0.1375

**Table 3 pharmaceutics-14-01741-t003:** The Py values of R1–R9 powder.

Sample	R1	R2	R3	R4	R5	R6	R7	R8	R9
Py (MPa)	139.3	141.5	144.1	136.9	132.2	132.6	124.1	127.3	135.5

**Table 4 pharmaceutics-14-01741-t004:** The energy of the different compression pressures.

Mean pressure(kN)	4.33	9.55	14.17	21.82	25.94	30.73	37.29	41.26	45.17
Compressive work (J)	2.222	4.405	5.936	8.054	8.726	9.538	11.023	11.675	12.044
Elastic work (J)	0.060	0.114	0.124	0.240	0.339	0.517	0.940	1.179	1.624
Plastic work (J)	2.182	4.325	5.871	7.843	8.404	9.405	10.092	10.535	10.341

**Table 5 pharmaceutics-14-01741-t005:** The relationship between compression speed and real compression time.

Speed (Tab/h)	Real Compression Time (ms)	
	Compression Time	Decompression Time
600	216	61
1200	92	34
1800	53	25
2400	35	20
3000	33	14
3600	32	11
4200	32	10
4800	32	8

**Table 6 pharmaceutics-14-01741-t006:** Effect of Venturi and Ring pressures during the mill process on HCTZ particles/mixed powders/tablets.

Parameter of Mill	D_50_ (μm)	D_90_ (μm)	ab	f	Py (MPa)	Cumulative Release	
						10 min (%)	20 min (%)
untreated	92.50	233.6	0.0578	0.1217	135.9	38.78	74.48
<100 mesh	77.66	192.7	-	-	-	-	-
Venturi 3 bar, Ring 2 bar	35.54	124.8	0.0585	0.1308	136.3	43.18	81.38
Venturi 3 bar, Ring 2.5 bar	25.78	95.13	-	-	-	-	-
Venturi 3.5 bar, Ring 2.5 bar	19.74	98.86	0.0624	0.1317	139.8	49.51	81.82
Venturi 3.5 bar, Ring 3 bar	14.37	80.69	0.0746	0.1528	149.5	49.63	82.85
Venturi 4 bar, Ring 3 bar	9.895	50.82	-	-	-	-	-

## Data Availability

Not applicable.

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
