# Peer review of "Hydrochlorothiazide/Losartan Potassium Tablet Prepared by Direct Compression"

_pharmaceutics, 2022, doi:10.3390/pharmaceutics14081741_

Round 1
Reviewer 1 Report
The paper is worthy of investigation. However, there are several point that need to be addressed before publication.
1. The quality fo the figures. Some of them barely are visible. Please, redraw the figures and make the legends clear and all numbers inside visible to the readers.
2. It may be worhty to enlarge the figures or separate them to enhance visibility.
3. It is not clear the D-optimal designed utilised in this work, have you included replicates at the central point? How can you calculate the intrinsic variability otherwise?
4. Standard devition values are missing in some fo the figures.
5. Data in fugre 9 is difficult to compare as the axis are diferent, Can be plotted in the same graph all the results and graph to be enlarged?
6. Statsitical analysis shoudl be performed, not only in the D-Optimal designed but also in the other figures such as dissolution, Pk data and so on.
7. The disucssion is very limited. How your data compares to other obtained by other authors? Can this work to be translated to other APIs? Cna you correlate the physicochemicla performance of the tablets with the dissolution behaviour? Check this paper: Predicting the critical quality attributes of ibuprofen tablets via modelling of process parameters for roller compaction and tabletting
Reviewer 2 Report
The manuscript presents an interesting study of the development of formulation, which is problematic due to physical-chemical properties of active pharmaceutical ingredients. From this point of view, it could be interesting for pharmaceutical technologists dealing with solid oral drug delivery systems.
Prior publication I would recommend to consider following changes:
1. To discuss the relation between parameters of tablets with the results of drug dissolution.
2. To extend discussion of the results of in vivo studies and to combine the results with observations from drug dissolution experiment.
3. Technical questions: the figures should be unified in terms of font size, some are difficult to read (e.g. Fig 8, and 9).
Reviewer 3 Report
The authors reported a study where they analyzed pharmaceutical formuşations based on hydrochlorothiazide (HCTZ)/losartan potassium (LOS-K) via compression and mechanical properties. The authors provided data and statistics to prove their outcome and they selected an optimal formulation.
1. The authors presented their purpose of the study; however, they did not state clearly in what way their work will contribute to the advances in the topic and a sort of novelty is needed to be highlighted.
2. Tab.1 Design for mixture experiment: The authors summarized a list of 9 experiments (R1-R9);however, the difference between them was not clearly stated. The authors are requested to clearly present their designs; additionally, the software or the methodology they have performed the experimental design should be mentioned also.
3. The authors mentioned only "lactose monohydrates and magnesium stearate" for their formulation's processing; the authors may consider also other additives and the advantage of lactose monohydrates and magnesium stearate
4. Fig 8 presented was not clear enough; the authors are requested to improve the quality and to explain better the differences between the presented graphs.
Round 2
Reviewer 2 Report
The current version of the manuscript is appropriate for publication.
Reviewer 3 Report
The authors have answered the addressed queries and updated the manuscript accordingly.